# Bisretinoids of the Retina: Photo-Oxidation, Iron-Catalyzed Oxidation, and Disease Consequences

**DOI:** 10.3390/antiox10091382

**Published:** 2021-08-29

**Authors:** Hye Jin Kim, Diego Montenegro, Jin Zhao, Janet R. Sparrow

**Affiliations:** 1Department of Ophthalmology, Columbia University Medical Center, 635 W., 165th Str., New York, NY 10032, USA; hk2798@cumc.columbia.edu (H.J.K.); dm3360@cumc.columbia.edu (D.M.); jz120@cumc.columbia.edu (J.Z.); 2Department of Pathology and Cell Biology, Columbia University Medical Center, 635 W., 165th Str., New York, NY 10032, USA

**Keywords:** retina, retinal pigment epithelium, bisretinoid lipofuscin, photosensitization, photo-oxidation, photodegradation, vitamin A-aldehyde

## Abstract

The retina and, in particular, retinal pigment epithelial cells are unusual for being encumbered by exposure to visible light, while being oxygen-rich, and also amassing photoreactive molecules. These fluorophores (bisretinoids) are generated as a byproduct of the activity of vitamin A aldehyde—the chromophore necessary for vision. Bisretinoids form in photoreceptor cells due to random reactions of two molecules of vitamin A aldehyde with phosphatidylethanolamine; bisretinoids are subsequently transferred to retinal pigment epithelial (RPE) cells, where they accumulate in the lysosomal compartment with age. Bisretinoids can generate reactive oxygen species by both energy and electron transfer, and they become photo-oxidized and photolyzed in the process. While these fluorescent molecules are accrued by RPE cells of all healthy eyes, they are also implicated in retinal disease.

## 1. Introduction

In the cell, reactive oxygen species (ROS) are generated due to the ability of molecular oxygen to accept electrons. ROS can be generated via both enzymatic and non-enzymatic processes. For instance, superoxide anions (O_2_^•−^) can be produced by the leakage of electrons from the electron transport chain during oxidative phosphorylation in the mitochondria, or by electron transfer from cytosolic nicotinamide adenine dinucleotide phosphate oxidase (NADPH-oxidase), as occurs in monocytes [1]. Hydrogen peroxide (H_2_O_2_), on the other hand, can be produced by the dismutation of superoxides, while hydroxyl radicals (^•^OH) are generated from hydrogen peroxide by single-electron transfer in the presence of Fe^2+^. ^•^OH is highly reactive toward most organic molecules.

Photosensitizers, both natural and synthetic, are molecules that can be activated by specific wavelengths of light, because they exhibit the conjugated double-bond systems that are necessary for intersystem crossing to the triplet-excited state, wherein the spin of the excited electron is reversed such that it is parallel to a ground-state electron. These π–electron systems also permit long-lived triplet states that allow sufficient time for reaction with neighboring target molecules [2,3]. The excited sensitizer in the triplet state may then transfer energy to molecular oxygen, thereby generating excited-state singlet oxygen (^1^O_2_), while the photosensitizer itself is returned to the ground state [1]. Subsequently, the singlet oxygen produced by illumination can attack the photosensitizer (photobleaching) and/or react with other molecules. Alternatively, a photosensitizer may transfer one electron to oxygen to produce a superoxide anion (O_2_^•−^). The latter can then be converted to hydrogen peroxide (H_2_O_2_) by superoxide dismutase [4], with H_2_O_2_ forming the highly reactive hydroxyl radical (^•^OH) in the presence of iron (Fenton reaction). The retina is unique in being burdened not only by exposure to light, but also by the age-related accumulation of photoreactive molecules; the latter form as a byproduct of the activity of vitamin A aldehyde—the chromophore necessary for vision.

## 2. Vitamin A Aldehyde and Bisretinoid Fluorophores

Bisretinoid fluorophores (Figure 1), which constitute the lipofuscin of the retina, form non-enzymatically in photoreceptors’ outer segments due to random reactions of vitamin A aldehyde with ethanolamine-containing phospholipids [5,6]. These fluorophores accumulate throughout the lifetime of an individual but, as discussed later, they undergo increased synthesis in some retinal disorders. The heterogeneous mixture of these fluorophores includes A2-GPE (A2-glycero-phosphoethanolamine) (molecular weight, 747.0), A2-DHP-PE (A2-dihydropyridine-phosphatidylethanolamine) (molecular weight, 1224.8 with dipalmitoyl PE (C16:0)), all-*trans*-retinal dimer-PE (all-*trans*-retinal dimer phosphatidylethanolamine) (molecular weight 1225.8 with dipalmitoyl PE (C16:0), and A2E (molecular weight, 592.9) [7,8,9,10,11,12,13,14]. Because these fluorophores accumulate in the retinal pigment epithelium (RPE) after formation in photoreceptor cells, they are typically measurable in the RPE but present at low levels or non-detectable in the neural retina.

Conversely, the phosphatidylpyridinium bisretinoid A2PE is measurable in photoreceptor cells or whole neural retinae, and is the immediate precursor of A2E [16]. The most recently characterized bisretinoid is alkyl-ether-lysoA2PE (1-alkyl ether-2-lysoA2PE) (1-octadecyl-2-lyso-*sn*-glyceroA2PE)—a fluorophore presenting with a single alkyl chain at the *sn*-1 position (340 and 440 nm, molecular weight, 999.5) [12]; this bisretinoid forms via the reaction of two vitamin A aldehydes with the ethanolamine head group of a glycerophospholipid with an ether bond at the *sn*-1 position, rather than the more common ester linkage. The structures of all of these pigments have been determined using various combinations of high-performance liquid chromatography, mass spectrometry, Fourier-transform infrared spectroscopy, nuclear magnetic resonance, and total synthesis [8,10,11,12,13,17]. All of the members of the bisretinoid family discussed above have been detected in both human and murine retinae. A2 in the various nomenclatures reflects formation from two vitamin A aldehyde molecules. Oxidative mechanisms are not involved in the formation of bisretinoid lipofuscin, unlike assumptions made for other forms of lipofuscin [18]; however, as will be discussed below, oxygen participates in the photolysis of these compounds.

Structural features shared amongst bisretinoid chromophores include the alternating single and double carbon–carbon bonds that originate from an aromatic head group. These bonds extend along the two side-arms of the molecule and into the terminal β-ionone rings (Figure 1). Each of these extended conjugation systems constitutes a chromophore, and allows for electron transitions to excited states after absorption of specific wavelengths of light. A2E and its isomers [8,16] A2-GPE [10] and A2PE [16] are characterized by a central pyridinium ring that houses a quaternary amine nitrogen. The nitrogen does not undergo deprotonation [8,19], and the positive charge on the pyridinium nitrogen is neutralized by a counterion (probably chloride). While the double bonds along the side-arms of A2E assume the *trans* (*E*) position, *Z*-isomers of A2E have double bonds at the C13/14 (isoA2E), C9/9′-10/10′, and C11/11′-12/12′ positions [8]. Highly excited states enable intramolecular modifications that produce these photoisomers. Rather than having a pyridinium ring, A2-DHP-PE presents with a non-charged dihydropyridine ring at its core [13], while the bisretinoid all-*trans*-retinal dimer presents with a cyclohexadiene ring. All-*trans*-retinal dimer-E and all-*trans*-retinal dimer-PE are all-*trans*-retinal dimers attached to PE via an imine function group (–C=N–), with a protonation state that is pH-dependent [7].

It is often stated that RPE lipofuscin consists of lipids and proteins [20]. However, proteomics analysis of lipofuscin-containing organelles purified from human RPE reveals that amino acids account for less than 2% of the constituents, yet these preparations express photoreactivity and the expected content of bisretinoids such as A2E, isoA2E, and all-*trans*-retinal dimer-phosphatidylethanolamine [21]. The compounds accounting for autofluorescence accumulation in cultured RPE cells after 2 weeks of feeding their isolated outer segments are not described [20]. Moreover, products of lipid peroxidation (4-hydroxynonenal, HNE; malondialdehyde, MDA) that are assumed to originate in the polyunsaturated-fatty-acid-rich outer segments of photoreceptors are also detected in RPE lipofuscin [21,22] (Figure 2). While these lipid fragments are not the primary constituents of RPE lipofuscin [23,24,25], photoreactive processes incited by bisretinoids may be involved in their production. A mechanism such as this is consistent with studies demonstrating that cell membranes incubated in the presence of light and isolated lipofuscin granules undergo lipid peroxidation [26,27,28].

## 3. Bisretinoids: Photosensitizers Unique to the Retina

It is often stated that because the retina is exposed to light and high oxygen tensions, its tissue is subject to oxidative stress [29]. A co-requisite to these conditions, however, is a chromophore capable of absorbing photons of specific energy via electron excitation. In the retina, bisretinoids of lipofuscin serve in this role, because their structures include systems of conjugated double bonds containing delocalized electrons that can absorb light and allow the bisretinoid to enter an excited singlet state. Thus, using cholesterol peroxidation assays, oxygen uptake [25,29,30], and ESR (electron spin resonance) spectroscopy, whole lipofuscin extracts, isolated lipofuscin-storage organelles, and suspensions of RPE cells have all been shown to exhibit photoreactivity under aerobic conditions, with efficiency being greatest using short-wavelength visible light [26,30,31,32,33,34,35,36,37]. Photoexcited pigments in lipofuscin granules have also been reported to form triplet states, with both singlet oxygen and superoxide anions being produced [25,26,31,38].

Synthesized bisretinoids such as A2E, A2-GPE, and all-*trans*-retinal dimer have also been shown to initiate photosensitization reactions that generate superoxide anion radicals (O_2_^•−^) and singlet oxygen (^1^O_2_) [10,11,34,35,37,38,39,40,41,42,43,44] (Figure 3). These photosensitization studies included the detection of a characteristic 1270 nm phosphorescence indicative of singlet oxygen, when A2E was irradiated at 430 nm [38,45].

Irradiation of the bisretinoid A2E with 430 nm light in the presence of DMPO (5,5′-dimethyl-1-pyrroline-N-oxide)—a radical trap used to detect oxygen-centered free radicals—also led to the appearance of an electron paramagnetic resonance (EPR) spectrum characteristic of DMPO-OH [46]. This EPR signal was inhibitable by superoxide dismutase [46]—an enzyme that converts superoxide anion radicals (O_2_^•−^) to molecular oxygen and hydrogen peroxide (H_2_O_2_) [4]. These findings were indicative of hydroxyl radical (^•^OH) formation, either directly, or following initial spin trapping of superoxide anions by DMPO. An increase in dihydroethidium (HEt) fluorescence and luminol-based chemiluminescence was also observed—both of which, on the basis of inhibition by superoxide dismutase, were indicative of superoxide anion generation by the transfer of one electron to oxygen when A2E was irradiated at 430 nm in cell-free systems [46].

The photoreactivity of bisretinoids is further manifest in cell-based studies. For instance, lipofuscin-free cultured RPE cells that were allowed to accumulate A2E [47,48], as well as cultured RPE cells that phagocytose lipofuscin-filled organelles [49], exhibited considerable phototoxicity to short-wavelength light in vitro. The death of blue-light-irradiated A2E-containing cells could also be blocked by oxygen-depleted media, but was increased in deuterium-based media, and did not occur if the cells were A2E-free [40]. Loss of viability was also decreased in the presence of histidine, 1,4-diazabicyclooctane (DABCO), and azide, all of which are efficient scavengers of singlet oxygen. On the other hand, an inhibitor of catalase (3-AT) and a scavenger of hydroxyl radicals and superoxides (DMTU) had no effect on the frequency of cell death following blue-light illumination of A2E.

Not surprisingly, measurements of the efficiency of singlet oxygen generation by A2E have led to the suggestion that A2E exhibits lower singlet oxygen production than photosensitizers used in photodynamic therapy. However, the detectability of singlet oxygen in these studies is complicated by the ease with which singlet oxygen reacts chemically with the parent bisretinoid molecule (discussed below). Moreover, the damaging effects of bisretinoid photosensitization are likely compounded by the photoproducts generated when A2E is photo-oxidized [50,51,52].

## 4. Photo-Oxidation and Cleavage of Bisretinoids

At any given time, measurement of bisretinoids—either chromatographically or via short-wavelength fundus autofluorescence—reflects a balance between synthesis and deposition in the RPE versus photodegradative processes. This is because bisretinoids not only generate reactive oxygen species by both energy and electron transfer, they also chemically quench these species and are simultaneously consumed. Photo-oxidation of bisretinoids has been recognized in mass spectra by the generation of a series of higher molecular mass derivatives [40] (Figure 3). The sizes of these species increased by increments of mass 16 (the molecular weight of oxygen). By chromatography, these peaks elute at retention times that are indicative of greater hydrophilicity. The generation of these photoproducts is accompanied by the loss of bisretinoids, the latter being diminished, however, when illumination is performed after oxygen depletion and in the presence of a singlet oxygen quencher.

A2E photo-oxidation is potentiated in deuterium oxide (D_2_O) [38,39]—a solvent that extends the lifetime of singlet oxygen—while the singlet oxygen quencher 1,2,2,6,6-pentamethyl-4-piperidinol attenuates A2E photo-oxidation [40]. Additionally, singlet oxygen generated by thermal decomposition of the 1,4-endoperoxide of 1,4-dimethylnaphthalene can substitute for blue light in mediating A2E oxidation [38,39,53], and the same species of oxygen-containing moieties form as with exposure to short-wavelength visible light [38,39]. Together, these findings are indicative of photo-oxidation of bisretinoids.

The oxygen-containing moieties formed by the photosensitization and oxidation of A2E and all-*trans*-retinal dimer, for instance, consist of three membered rings that include one oxygen atom (epoxide; C-O-C; epoxide-A2E), heterocyclic rings of four carbons and one oxygen (furan) (furano-A2E), and heterocyclic rings that include four carbons and an endoperoxide (O–O) (peroxy-A2E) [11,15,38,39,54] (Figure 3). Molecular fragmentation readily occurs at positions of oxygen addition; as a result, aldehyde- and carbonyl-bearing degradation products are released (Figure 4). These low-molecular-weight products include methylglyoxal and glyoxal [55,56]—known mediators of advanced glycation end product (AGE) formation [57,58]. Ketones and aldehydes are relatively long lived and, therefore, can diffuse from their site of origin to reach and attack other targets intracellularly or extracellularly. These reactive dicarbonyls modify molecular structure and function by forming adducts with proteins, phospholipids, and nucleotides. Cleavage of oxidized A2E, with diffusion of fragments, explains the observation that when fibronectin was used as a substrate for A2E-containing RPE cells, irradiation elicited A2E photodegradation, and the fibronectin became AGE-modified [59,60]. The photodegradation products released from bisretinoids can cross-link proteins and promote resistance to the activity of matrix metalloproteinases [61]. Proteins similarly modified by dicarbonyls have been described in drusen [62,63]—the sub-RPE deposits associated with AMD [64].

The finding that complements can be activated in serum overlying irradiated, A2E-laden RPE cells is also consistent with the view that reactive cleavage products of A2E are generated by photo-oxidation [65]. These oxidized forms of bisretinoid are detectable in human and murine RPE cells [39], and these bisretinoid-photodegradative processes are responsible for the lower bisretinoid levels measured in albino versus black mice and in light- versus dark-reared mice [66]. Similarly, these processes may be the reason why early and intermediate AMD are associated with lower fundus autofluorescence intensity—measured as quantitative fundus autofluorescence, (qAF)—in the central retina [67]. Methylglyoxal (MG) is known to be generated as a byproduct of metabolic pathways such as glycolysis [57,68]. The release of MG via the photodegradation of bisretinoids reflects a previously unrecognized source.

## 5. Bisretinoid Fluorescence

All of the known bisretinoids of RPE lipofuscin are fluorescent compounds. Each hydrophobic retinoid-derived arm of a bisretinoid molecule constitutes a system of double-bond conjugations, and each serves as a chromophore—one arm absorbing from the visible region of the spectrum, and the other arm absorbing ultraviolet wavelengths (Figure 1). The absorbance wavelengths are determined by the lengths of the systems of alternating double and single bonds. A2E, A2-GPE, and alkyl-ether-lysoA2PE [10,12,16], for example, have an absorbance maximum (λ_max_) in the visible spectrum at ∼440 nm that is generated within the long arm, and an absorbance at ∼340 nm that can be assigned to the short arm. The absorbance/excitation spectrum of A2E/isoA2E is narrower than that of whole lipofuscin [69]. The absorbance maxima of the other bisretinoid pigments are: all-*trans*-retinal dimer, λ_max_∼290, 432 nm; all-*trans*-retinal dimer-E/all-*trans*-retinal dimer-PE, λ_max_ ∼290, 510 nm; and A2-DHP-PE, λ_max_∼333, 490 nm. The additional redshift to 510 nm absorbance in the case of all-*trans*-retinal dimer-PE and all-*trans*-retinal dimer-E occurs due to the protonation of the Schiff base nitrogen [7,11]. As noted above, some of the bisretinoids retain a phospholipid moiety, but this portion of the molecule does not make a contribution to absorbance at wavelengths greater than 250 nm.

Despite the variability in excitation maxima amongst bisretinoid species, the emission maxima of bisretinoids are similar. With excitation at 430 nm, A2E and iso-A2E have relatively broad emission spectra, with maxima at 600 nm [19] and orange fluorescence. Excitation of all-*trans*-retinal dimer-PE and all-*trans*-retinal dimer-E at their approximate absorbance maxima (~500 nm) also produces an emission centered at 600 nm, but its fluorescence intensity is weaker and the maxima are less clearly defined. Unconjugated all-*trans*-retinal dimer presents with greater fluorescence emission than A2E and all-*trans*-retinal dimer-PE when excited at 430 nm, but peak emission occurs at 510 nm and the spectral width is reduced. When considered relative to the corresponding absorbance, A2-DHP-PE also exhibits fluorescence (emission monitored at 630 nm; excitation at 430 nm) of greater intensity than A2E and all-*trans*-retinal dimer-PE [13]. The combined emission spectra of A2E/isoA2E and all-*trans*-retinal dimer-PE/all-*trans*-retinal dimer-E correspond well to the emission maximum (590–620 nm) of native RPE lipofuscin. Absorbances in the visible spectrum are significant, since these wavelengths reach the retina.

With excitation of individual bisretinoid chromophores and whole lipofuscin, and with excitations that elicit in vivo fundus autofluorescence [10,69,70], maximum fluorescence emission undergoes excitation-dependent hyperchromic and hypochromic shifts; the emission exhibits redshifts (hyperchromic change) in response to longer wavelengths of excitation; this shift is indicative of a complex mixture of bisretinoids.

Moreover, the wavelength of emission maxima and emission intensity is also influenced by the surrounding milieu [10,12,71]. Specifically, emission maxima occur at shorter wavelengths and higher intensities when associated with relatively more hydrophobic environments, while in more polar solvents the emission is redshifted. When introduced to TMAC (trialkyl-methylammonium chloride)—a positively charged detergent with which A2E would not associate—fluorescence emission was minimized [72].

Finally, fluorescence intensities of some oxidized forms of bisretinoid can be many times greater than the parent molecule. For instance, addition of one and two oxygen atoms on the short arm of A2E increases fluorescence efficiency by as much as 12-fold [71]. This property is consistent with an increase in overall fluorescence emitted from a bisretinoid mixture, despite a reduction in the amount of each parent molecule.

The depletion of fluorescence associated with oxidation of bisretinoid [73] is typical of photobleaching (Figure 5). Fluorescence quenching of bisretinoid has been observed in vivo as a reduction in fundus autofluorescence [74,75]. Specifically, in non-human primates during adaptive optics scanning laser ophthalmoscopy (AOSLO) with in vivo fluorescence capability (568 nm excitation), the natural autofluorescence of the RPE was found to be reduced immediately after irradiation. At lower light exposures, the fluorescence bleaching can recover fully. These photobleaching processes have been replicated in cell-based and non-cellular assays [73]. For instance, cell-associated in vitro modeling of A2E fluorescence bleaching has shown that the process involves photo-oxidation and photodegradation of bisretinoids, the latter being measured as a loss of specific absorbance [73]. The potential for autofluorescence recovery is dependent on light dose and antioxidant status.

## 6. Iron-Assisted Oxidation of Bisretinoids

Since iron is able to accept and donate electrons, this metal, in its ferrous state (Fe^2+^), can react with H_2_O_2_, yielding highly reactive hydroxyl radicals (^•^OH) (Fenton reaction) [1]. Just as with photomediated oxidation [55,56], the conjugated systems of double bonds within bisretinoid structures can be oxidized as a consequence of iron-mediated hydroxyl radical production [76]. The latter mechanism explains the reduction in HPLC-(high-performance liquid chromatography) quantified bisretinoid levels in liver-specific hepcidin (Hepc)-knockout (*LS-Hepc*^−/−^) mice; these mice present with elevated iron in their blood and increased free (labile) iron levels in their retinal and RPE cells [77,78]. Similarly, the excessive iron in the RPE cells of mice deficient in ceruloplasmin (Cp) and hephaestin (Heph) (*Cp*^−/−^; *Heph*^−/−^ mice) [79] is associated with toxic bisretinoid photo-oxidation and degradation [76].

Conversely, the increase in HPLC-quantified bisretinoids and SW-AF (short-wavelength autofluorescence) in mice treated with the iron chelator deferiprone (DFP) is indicative of DFP-mediated reduction in the endogenous iron-associated degradation of bisretinoids [76]. The FDA-(Food and Drug Administration) approved iron chelator deferiprone (DFP) is orally absorbed and cell-permeable, and is known to reduce serum iron and intracellular iron levels in the retina [76,80,81]. DFP not only binds iron, it also oxidizes Fe^2+^ to Fe^3+^, thus impeding the effects of Fe^2+^—the major catalyst of free radical damage to cells [82].

The production of hydroxyl radicals via the Fenton reaction depends on continued availability of Fe^2+^ (ferrous iron). Nevertheless, light may overcome this limitation. Thus, in in vitro experiments, it was observed that the oxidation of A2E was greater when iron, H_2_O_2_, and light were provided in the reaction mixture, as compared to H_2_O_2_ and Fe^2+^ (in darkness) or light alone. Under these conditions, it was surmised that light in the presence of the photosensitizer A2E potentiated the Fenton reaction due to reduction of Fe^3+^ to Fe^2+^ (one electron transfer) by superoxides (O_2_^•−^) [46,76,83,84] (Figure 6). Synergy between the Fenton reaction, photosensitization of bisretinoids, and redox conversion of the Fe^2+^ and Fe^3+^ oxidation states is significant to the retina since, unlike most tissues, the retina is exposed to visible light [77].

## 7. Antioxidant Protection

The tripeptide glutathione (L-γ-glutamyl-L-cysteinyl-glycine, GSH) is present in cells in millimolar concentrations [85]. By employing colorimetric assays, chromatography, and mass spectrometry, it was demonstrated that GSH can donate hydrogen atoms to—and form conjugates with—photo-oxidized forms of the bisretinoids A2E and all-*trans*-retinal dimer, as well as with their photocleavage products [86]. As expected based on its detoxification capabilities [85], GSH formed an adduct with methylglyoxal both non-catalytically and by glutathione-S-transferase (GST)-mediation. The chemical reduction by GSH involved the donation of a hydrogen atom from each of two GSHs, and the ratio of GSH consumed to GSSG formed was consistent with GSH being utilized for both reduction and adduct formation. The adducts preferentially formed with photo-oxidized forms of A2E carrying two or more oxygen atoms. The binding of GSH to these photocleavage products of A2E likely serves to limit their reactivity.

Using in vitro studies designed to detect cellular damage together with A2E as a model bisretinoid, vitamin E has been shown to attenuate 430 nm light-induced oxidative injury in cells that accumulated A2E [87]. By fast atom bombardment (FAB) mass spectrometry, the protection was associated with reduced oxidation of A2E. In mice receiving a diet supplemented with vitamin E, HPLC analysis revealed higher levels of RPE bisretinoid indicative of reduced oxidative loss of the fluorophores [66]. In *Abca4*^−/−^ mice known to exhibit photoreceptor cell loss by 8 months of age, outer nuclear layer thinning was alleviated. These protective effects reflect the ability of vitamin E to serve as an electron donor that prevents bisretinoids from being oxidized.

Four phytochemicals—bilberry-derived anthocyanins [88], resveratrol [87], the phase 2 inducer sulforaphane [89], and quercetin [90]—also confer resistance to photo-oxidative processes initiated in RPE cells by bisretinoids. The protective effect of anthocyanins is enabled both by the unsaturated diene conjugation in the C (pyrane) ring that allows for singlet oxygen quenching and by the presence of hydrogen-donating hydroxyl groups on the B ring. Quercetin also defends RPE cells against light damage in vitro by preventing the photo-oxidation and photodegradation of bisretinoids. Sulforaphane itself does not directly partake in antioxidant processes, but instead acts indirectly to increase antioxidant defense through the induction of cellular enzymes such as glutathione-S-transferases (GST), NAD(P)H:quinone reductase (NQO1), epoxide hydrolase, γ-glutamylcysteine synthetase, and UDP-glucuronosyl-transferases [91].

## 8. Disease Significance

The disorder best known for being characterized by elevated bisretinoid formation and accumulation is recessive Stargardt disease (STGD1) [92,93]. In STGD1, handling of retinaldehyde is impaired due to pathogenic variants in the ATP-binding cassette (ABC) subfamily A member 4 (ABCA4) transporter expressed by photoreceptor cells—both rods and cones. Consequently, the tendency of retinaldehyde to form adducts non-enzymatically is accelerated. STGD1 is characterized by progressive loss of central vision, usually beginning in the first two decades of life. Flecks presenting as hyperautofluorescence foci that exceed the overall levels of fundus autofluorescence are also a feature of STGD1. The intense SW-AF signal associated with fundus flecks has been shown to originate from augmented lipofuscin formation in impaired photoreceptor cells [94]. In some patients exhibiting retinal degeneration with a pattern dystrophy phenotype, while being negative for *ABCA4* mutations, qAF is also increased, suggesting that elevated qAF may be an ancillary feature of the disease pathogenesis [95].

Analysis by qAF has revealed that increases in SW-AF indicative of elevated bisretinoid lipofuscin formation can also be detected as secondary features in other diseases. For instance, in acute zonal occult outer retinopathy (AZOOR), SW-AF intensity is elevated at the border between the diseased and non-diseased retina (AZOOR line). This elevation coincides with disruptions of the photoreceptor-attributable reflectivity layers in spectral domain optical coherence tomographic scans [96]. In Best vitelliform macular dystrophy, qAF levels are elevated many-fold within the lesion, wherein outer nuclear thinning is also observed [97]. Similarly, in retinitis pigmentosa (RP), qAF in the ring of hyperautofluorescence that is often visible in the fundus exhibits intensity levels that are actually elevated relative to the same fundus location in healthy eyes [98]. Within these rings, structural and functional studies attest to photoreceptor cell degeneration [99,100,101,102,103]. Excessive production of bisretinoids is likely a secondary feature of the disease that is triggered by photoreceptor cell impairment. Given the phototoxic processes associated with bisretinoids, the increase in bisretinoid production may be significant to the disease process.

A link between bisretinoid accumulation and light-mediated photoreceptor cell degeneration is demonstrated by mice and rats deficient in the receptor tyrosine kinase Mer (*Mertk*^−/−^), which enables phagocytosis of photoreceptors’ outer segments by RPE cells [104,105,106]. The defective RPE-mediated phagocytosis leads to bisretinoid accumulation in the subretinal space [107]. Under these conditions, the photoreceptor cells are impaired, and bisretinoids measured by HPLC are elevated relative to wild-type mice. Moreover, the rate of photoreceptor cell degeneration is more rapid in albino *Mertk*^−/−^ mice/rats that also experience higher intraocular light, and the degeneration is more rapid in mice carrying the wild-type amino acid variant in Rpe65 (Rpe65-Leucine450), which yields greater bisretinoid formation than methionine in this position [108]. These findings are consistent with studies in *Abca4*^−/−^ mice reporting more pronounced light-mediated photoreceptor cell death in *Abca4*^−/−^ mice than in the wild-type mice; photoreceptor cell loss is also more pronounced in older than in younger mice [109].

Susceptibility to age-related macular degeneration (AMD) is related to multiple genetic and environmental factors [110,111]. The influence of non-genetic factors such as smoking [112,113] and nutritional status in relation to dietary and supplemental antioxidants [114,115,116,117,118,119] is widely considered to reflect the involvement of oxidative mechanisms in AMD pathogenesis. Furthermore, because antioxidants can protect against bisretinoid photo-oxidation [87,88,89] (Section 7), the beneficial effects of antioxidant intake in combatting AMD progression may be at least in part attributable to the suppression of photo-oxidative processes that precede bisretinoid photodegradation. In this regard, several studies and a meta-analysis have supported a link between AMD and sunlight exposure [120,121,122,123,124,125]. Interestingly, in studies of AMD cohorts [67], it was found that qAF in patients with soft and cuticular drusen were within the 95% confidence intervals of qAF values in age-similar healthy eyes. qAF levels were below the 95% confidence intervals in patients with reticular pseudodrusen. Decreased lipofuscin in AMD likely reflects impaired photoreceptor cells and/or bisretinoid lipofuscin photodegradation processes.

## 9. Conclusions

In addition to metabolic sources of reactive forms of oxygen, photo-oxidative processes initiated by RPE bisretinoid lipofuscin contribute to oxidative stress within the RPE. These bisretinoid fluorophores form due to non-enzymatic reactions of retinaldehyde in photoreceptor cells, and are transferred to the RPE in the phagocytosed outer segment membrane. While these compounds can be acted upon by the hydrolytic enzymes phospholipase D and phospholipase A2 at phosphodiester and fatty acid linkages, respectively, these compounds appear to be otherwise refractory to lysosomal enzyme degradation. Nevertheless, they are depleted by photo-oxidation and cleavage.

## Figures and Tables

**Figure 1 antioxidants-10-01382-f001:**
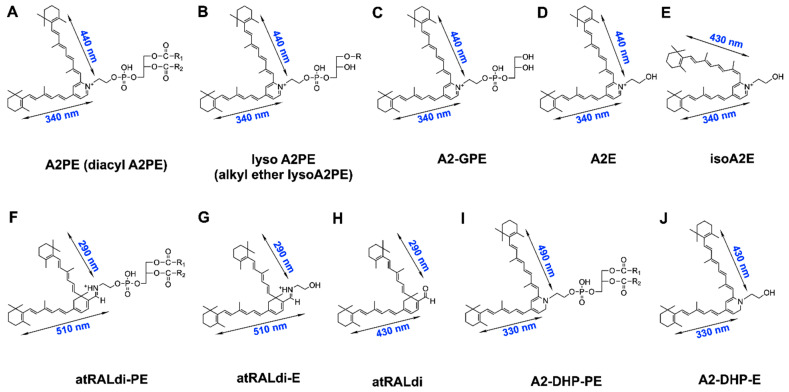
Structures of some of the bisretinoid fluorophores (**A**–**J**) known to be constituents of the lipofuscin of the retina. The fluorescence capability of bisretinoids is provided by the extensive system of carbon–carbon double bonds within the retinaldehyde-derived chromophore. Absorbance maxima (λ_max_) corresponding to the long and short arms of each bischromophore are indicated. Phosphate cleavage of A2PE generates A2E. R, R_1_, and R_2_ are fatty acids with various carbon numbers and multiple double bonds. Adapted from [15].

**Figure 2 antioxidants-10-01382-f002:**
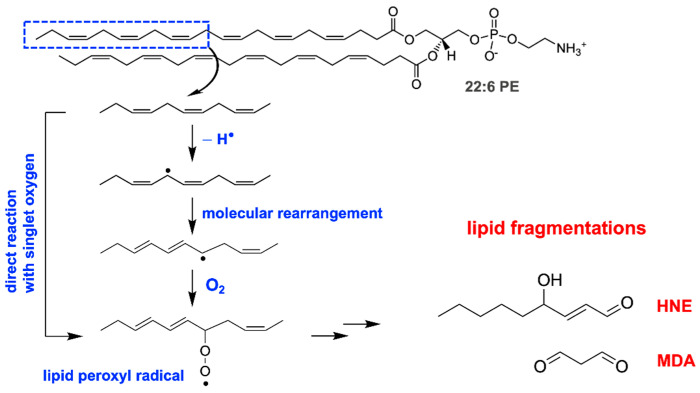
Photosensitization can initiate lipid peroxidation. Reaction with singlet oxygen leading to lipid peroxide formation in the hydrophobic interior of lipid membranes. 1,2-Didocosahexaenoyl-*sn*-glycero-3-phosphoethanolamine (22:6 PE) that contains polyunsaturated fatty acids is shown. Hydroxyl radicals also initiate lipid peroxidation via hydrogen atom abstraction from a methylene (-CH_2_-) group. Once peroxyl radicals are generated through multiple steps (stacked arrows), the peroxyl radicals cause the formation of aldehyde-bearing end products of lipid peroxidation (lipid fragmentation), such as HNE (4-hydroxynonenal) and MDA (malondialdehyde).

**Figure 3 antioxidants-10-01382-f003:**
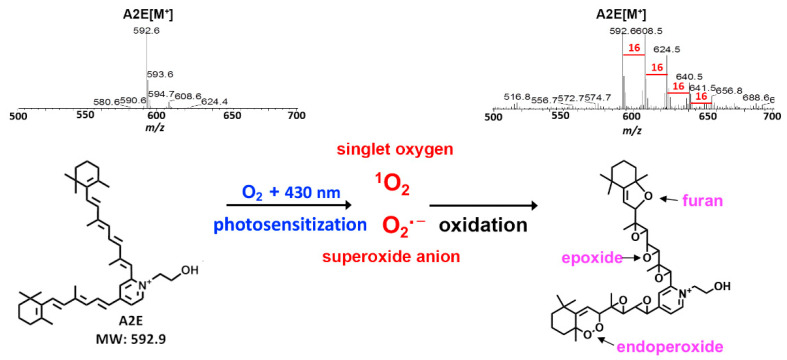
Photosensitization and photo-oxidation of bisretinoids illustrated using the bisretinoid fluorophore A2E as a model. The mass to charge ratio (*m/z*) of nonirradiated A2E is 592. Bisretinoids such as A2E initiate photosensitization reactions that generate singlet oxygen and superoxides. Singlet oxygen is inserted into double bonds along the side-arms of the molecule. The series of peaks from *m*/*z* 608 to 656 detected by electrospray ionization (ESI) mass spectroscopy indicate the addition of oxygen (mass 16), and reflect photo-oxidation at carbon–carbon double bonds. Proposed structures of the oxygen-containing modifications of A2E include furan, epoxide, and endoperoxide moieties.

**Figure 4 antioxidants-10-01382-f004:**
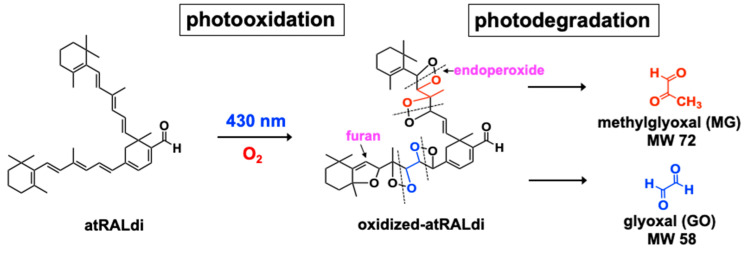
Bisretinoid photo-oxidation followed by photocleavage leads to the release of photodegradation products that include the dicarbonyls methylglyoxal (MG) and glyoxal (GO). MW: molecular weight. The bisretinoid all-*trans*-retinal dimer (atRALdi) serves to illustrate.

**Figure 5 antioxidants-10-01382-f005:**
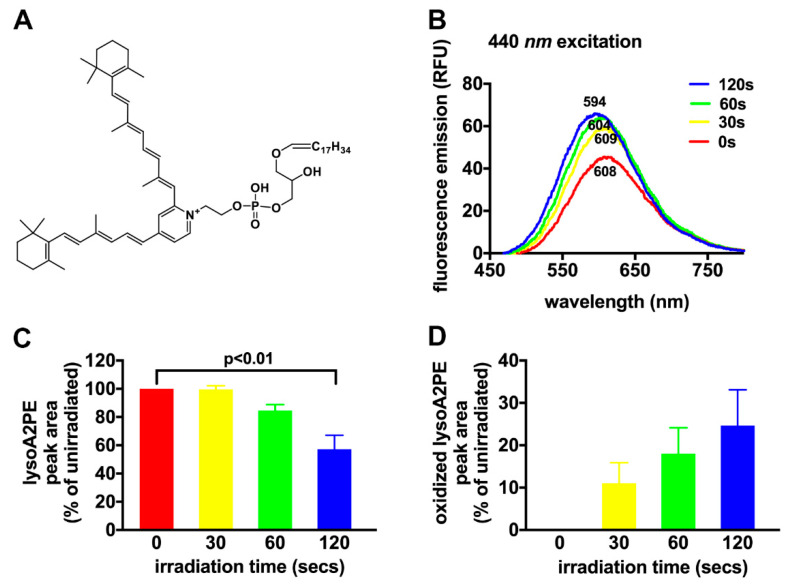
Photo-oxidation of alkenyl ether lysoA2PE (lysoA2phosphatidylethanolamine) (P-18:0/0:0) (P: plasmalogen, one *cis* double bond on the fatty acid chain). (**A**) The structure of lysoA2PE. (**B**) With excitation at 440 nm, lysoA2PE (in phosphate buffer with 2% DMSO (dimethylsulfoxide)) has an emission maximum at 608 nm. Fluorescence emission of lysoA2E when not irradiated (red trace) and when irradiated for the times indicated (yellow, green, and blue traces). Emission peak wavelengths are indicated adjacent to each trace. Note the increase in emission intensity and the peak shift to shorter wavelengths with increasing duration of irradiation. (**C**) Photo-oxidative loss of lysoA2PE is plotted as normalized peak area and as a function of irradiation time. (**D**) Formation of oxidized lysoA2PE is presented as normalized peak area and as a function of irradiation time.

**Figure 6 antioxidants-10-01382-f006:**
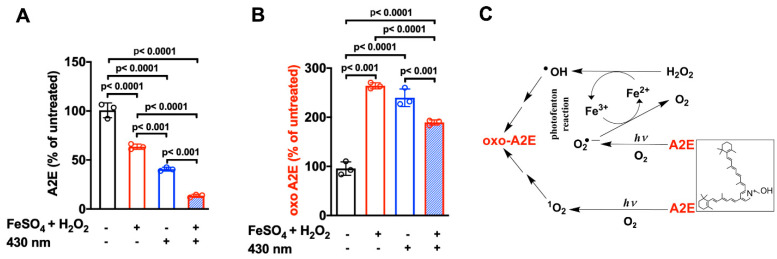
Synergistic effects of iron (Fe), hydrogen peroxide (H_2_O_2_), and light (*hv*) in the oxidation of the bisretinoid A2E. (**A**) Photo-Fenton reaction and (**B**) UPLC quantitation of A2E and oxidized A2E (oxo-A2E) after incubation with FeSO_4_ and H_2_O_2_, with and without 430 nm irradiation. Means ± SD, *n* = 3. *P*-values determined by two-way ANOVA and Tukey’s multiple comparison test. (**C**) Proposed mechanism: Hydroxyl radical (^•^OH) generated by Fe^2+^ and H_2_O_2_ (Fenton reaction) oxidizes A2E and produces Fe^3+^. Fe^2+^ is replenished by donation of an electron from the superoxide anion (O_2_^•−^); the latter forms via irradiation of A2E. Photosensitization of A2E generates O_2_^•−^ and singlet oxygen ^1^O_2_, which also oxidize A2E. Adapted from [77].

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
