# Peer review of "Bisretinoids of the Retina: Photo-Oxidation, Iron-Catalyzed Oxidation, and Disease Consequences"

_antioxidants, 2021, doi:10.3390/antiox10091382_

Round 1

Reviewer 1 Report

The review summerize the role of bisretinoid in photooxidation, iron-catalyzed oxidation and disease consequences. The topic is interesting, and the review is well-organized and informative. However, some figures need to be revised.

  1. In Figure 1, the color of  Ex. and Em. wavelength in the text was recommended to be shown with different colors.
  2. In Figure 1, the picture of each bisretinoid fluorophore was recommended to be labeled with A, B, C....
  3. In Figure 2, why they are two arrows at the bottom of the picture.
  4. Figure 3 should be revised to be more concise and logical.
  5. In Figure 5, more information should be provided in each subfigure, because it was hard to know the meaning of this data.

Author Response

1. In Figure 1, the color of Ex. and Em. wavelength in the text was recommended to be shown with different colors.

Response. Thank you. Colors have been added for the Ex and Em wavelengths.

2. In Figure 1, the picture of each bisretinoid fluorophore was recommended to be labeled with A, B, C...

Response. Thank you. We have added labels (A to J) to the figure.

3. In Figure 2, why they are two arrows at the bottom of the picture.

Response. The stacked arrows here represent multiple steps being involved for the final lipid fragmentation compounds.

Revision. Legend 2. ‘Once peroxyl radical is generated through multiple steps (stacked arrows)…’

4. Figure 3 should be revised to be more concise and logical.

Response. We have revised the Figure 3 as suggested and modified figure legend: ‘Photosensitization and photooxidation of bisretinoid illustrated using the bisretinoid fluorophore A2E as a model. The mass of unirradiated A2E is 592. Bisretinoids such as A2E initiate photosensitization reactions that generate singlet oxygen and superoxide. Singlet oxygen is inserted into double bonds along the side-arms of the molecule. The series of peaks from m/z 608 to 656 detected by electrospray ionization (ESI) mass spectroscopy indicate the addition of oxygen (mass 16) and reflect photooxidation at carbon-carbon double bonds. Proposed structures of the oxygen-containing modifications of A2E are furan, epoxide and endoperoxide moieties.’

5. In Figure 5, more information should be provided in each subfigure, because it was hard to know the meaning of this data.

Response. Thank you. We have added more information in the Figure 5 legend and revised the Figure 5 for better understanding. The following sentences have been added: (B) ‘Fluorescence emission of lysoA2E when not irradiated (red trace) and when irradiated for the times indicated (yellow, green and blue traces). Emission peak wavelengths are indicated adjacent to each trace. Note the increase in emission intensity and the peak shift to shorter wavelengths with increasing duration of irradiation.’ (C) Photooxidative loss of lysoA2PE is plotted as normalized peak area and as a function of irradiation time. (D) Formation of oxidized lysoA2PE is presented as normalized peak area and as a function of irradiation time.

Reviewer 2 Report

This is a seminal contribution to the retinal field and very important for clinicians because modern diagnostics advances to spectroscopy and related technics also for screening patients.

Minor point: are so many Keywords required?

Author Response

Minor point: are so many Keywords required?

Response. Thank you.

We have reduced keywords. The revised list of keywords is: retina; retinal pigment epithelium; bisretinoid lipofuscin; photosensitization; photooxidation; photodegradation; vitamin A-aldehyde.